# Recording vaccine doses administered: A global analysis of tally sheet design for infant and child immunizations

Ariel Higgins-Steele*, Stephanie Shendale , Jan Grevendonk , Marta Gacic-Dobo, M. Carolina Danovaro-Holliday

Department of Immunization, Vaccines and Biologicals, World Health Organization, Geneva, Switzerland

* higginssteelea@who.int

## Abstract

Most countries use tally sheets across immunization programmes for health workers to mark vaccine doses administered to every person, alongside home-based records, yet their design and use is poorly understood at the global level. This paper presents a multi-country analysis on routine childhood immunization tally sheet content and design sourced from the annual electronic WHO/UNICEF Joint Reporting Form on Immunization (eJRF), which collected these forms for the first time in 2024 (for 2023). Of the total submissions (N = 71) of tally sheets by reporting entities, 51 submissions met the criteria for inclusion as a valid tally sheet allowing for data extraction. The 51 tally sheets were from countries and territories across all six regions of the World Health Organization (WHO) and all four World Bank income classifications. Analysis showed heterogeneity in immunization tally sheet design, core data fields, and the extent to which the available tally sheets aligned with globally recommendations found in recent vaccine-specific and programmatic guidance related to catch-up vaccination. As national immunization programmes and vaccination schedules protect against more diseases, and thereby become more complex, fit-for-purpose tally sheet design and instructions as well as support mechanisms for health workers on tally sheet use are essential, similar to what is recommended for home-based records.

## Introduction

Tally sheets are the main recording tool for health workers to mark each time a vaccine dose is given at the point-of-delivery. They are also used to aggregate total doses administered, often monthly, but sometimes more frequently, to monitor performance against health facility targets. [1]. Tally sheets complement other immunization and primary health care recording and reporting tools such as vaccine stock management tracking and immunization registers and home-based records, often known as vaccination cards, where doses are also recorded each time an individual is vaccinated.

**Data availability statement:** Data related to the WHO/UNICEF Joint Reporting Form on Immunization can be found at: https://immunizationdata.who.int.

**Funding:** The authors received no specific funding for this work.

**Competing interests:** The authors have declared that no competing interests exist.

Global immunization guidance recommends tally sheets include specific data elements, e.g., neonatal tetanus protection at birth [2], hepatitis B birth dose timing (timely ≤24 hours or >24 hours) [3], and age segregation of on-time vaccine doses and late or delayed vaccine doses [4]. Beyond these recommendations, there is currently no global guidance specific to the design of tally sheets due to the heterogeneity of vaccination schedules and immunization data systems. Furthermore, though often just for immunization, some tally sheets may include other interventions such as vitamin A supplementation and antiparasitic drugs. Most supplementary immunization activities (vaccination campaigns) employ different tally sheets designed specifically for said campaign. Global guidance for home-based records [5,6] provides considerations on simple, user-friendly design, though it is for a different vaccination record and geared towards a caregiver.

A country updates its tally sheet and other recording and reporting tools each time a new vaccine or dose is introduced, or when any change in the recommended vaccination schedule occurs [7]. New vaccine introduction and renewed emphasis on making catch-up an integral part of immunization programmes, especially following the *Big Catch-up* initiative [8,9] provide opportunities for a country to review and revise its immunization tally sheet.

To accurately capture doses administered, a well-designed tally sheet includes date, vaccination clinic or post and session characteristics (e.g., fixed or outreach), some data about the person receiving the vaccine, and dose and vaccine information aligned to the national immunization schedule. These data fields should be displayed in a simple, user-friendly format with instructional elements for a health worker to mark doses administered during vaccination activities.

This paper presents a multi-country analysis on routine childhood immunization tally sheet design and content sourced from the annual electronic WHO/UNICEF Joint Reporting Form on Immunization (eJRF), which collected these forms for the first time in 2024 (for tally sheets used in 2023).

## Materials and methods

In 2023, as part of the annual electronic WHO/UNICEF Joint Reporting Form on Immunization (eJRF) process, all reporting entities (N = 214) were asked what type of system (paper, digital, or hybrid) was used to record routine vaccination doses. A total of 208 reporting entities submitted eJRF data in 2024 for the previous year. Those responding paper or hybrid to record routine doses were requested to submit a copy of their immunization tally sheet. The request was included within the systems indicators section of the eJRF, under paper-based systems, and worded as: "please upload a copy or image of the tally sheet and other aggregate/summary forms". Tally sheets used by the national immunization programme for routine immunization (inclusive of timely and delayed or catch-up vaccination), and not for vaccination campaigns, were requested. This was the first time that countries were asked to share tally sheet documents via the eJRF; this question was added in order to have more information to guide catch-up efforts and thus allowing for a cross-country analysis and comparison of submitted tally sheets. For this analysis, the inclusion criteria were having features

of a vaccination tally sheet for infant and child vaccination as part of the routine vaccination schedule and programme (i.e., fields to record individual routine vaccine doses given to infants and children, site and session characteristics, etc). Exclusion criteria were submissions with characteristics of aggregate reporting form formats (monthly or annual), tally sheets for supplementary immunization activities, tally sheets for adolescent and/or adult vaccination, and other record types (i.e., home-based records, vaccine stock reports).

Documents submitted by countries were reviewed by at least two of the authors and screened for inclusion. Document types excluded from the analysis included formats corresponding to reports (monthly, annual, or vaccine stock), home-based records, and tally sheets specifically for vaccines beyond childhood (e.g., HPV vaccination for adolescents).

Priority data elements of tally sheets were jointly identified by three members of the research team with expertise in immunization data systems and life-course strategies for vaccination. Tally sheets in English, French and Spanish were verified by members of the team with fluency in the language and assisted by Google Lens for other languages. Data was then extracted, cross-checked, and analyzed (S1 Data). Descriptive analysis provided the number and proportion of WHO reporting entities of included tally sheets, and results were compiled at the global and WHO regional levels, as well as by World Bank income classification (2024 revision) [10].

Of the total submissions (n = 71) under the category of tally sheets, 51 separate submissions from Member states and reporting entities – with no duplicates or geographic overlap – met the criteria for inclusion as a valid tally sheet sufficient for data extraction. Of the retained tally sheets, 59 were for WHO member states and 2 were from other reporting entities, i.e., territories; 20 were excluded as classified in Fig 1.

## Results

### Tally sheets

Among the 51 retained tally sheets, these are from all six WHO regions and all four World Bank income classifications [10] (see Table 1). In terms of language, 21 tally sheets retained are in English, 13 are in French, 5 are in Spanish and the remaining 12 were in other languages.

Three broad styles are observed (see Figs 2–4): tally by doses and age groups for which a vaccine is administered, contact points for doses of vaccine by recommended age of administration and age group given, and register-style line-listing children by name, date of birth, and date of vaccine doses administered.

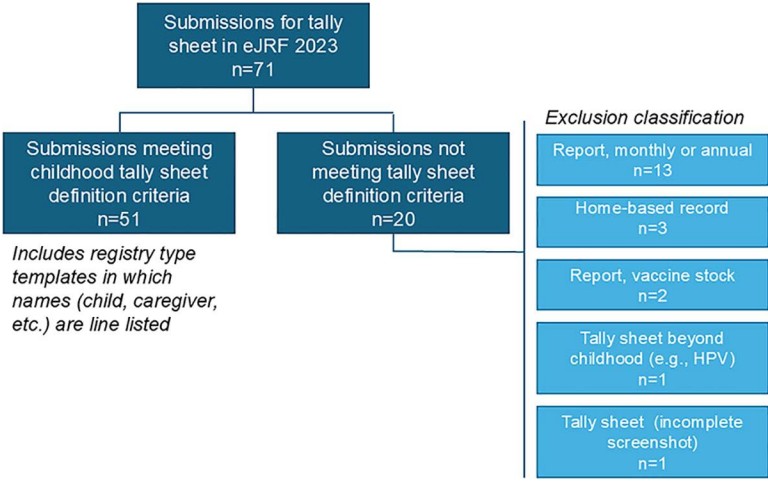

**Fig 1. Selection process for tally sheet analysis.**

**Table 1. Global eJRF tally sheet analysis, 2023 (data collected in 2024), by WHO region and World Bank income classification (n=51).**

| | AFR n=23 | AMR n=12 | EMR n=5 | Others[1] n=11 | Total n=51 |
|---|---|---|---|---|---|
| *Classification* | | | | | |
| High income | -- | 4 (33) | 1 (20) | 1 (9) | 6 (12) |
| Upper middle income | 2 (9) | 5 (42) | 1 (20) | 4 (36) | 12 (24) |
| Lower middle income | 13 (57) | 3 (25) | 2 (40) | 6 (55) | 24 (47) |
| Low income | 8 (35) | -- | 1 (20) | -- | 9 (18) |

[1]Others category includes countries from WHO European Region (EUR), South-East Asia Region (SEAR), and Western Pacific Region (WPR)

**Fig 2. Tally sheet design with doses grouped by vaccine type with age segregated groups.**

## Site and session characteristics

Core elements of immunization site and session characteristics include location or health facility name, date/timeframe, and service delivery modality. A location field to designate where doses are administered was observed in most tally sheets (n=44, 86%); some tally sheets included write-in section for subnational administrative areas whereas others simply had a field titled "location" or "health facility" with a line to fill in text. It is possible that electronically generated or booklet versions of the tally sheets with elements or fields not seen may have this information.

Date: _________ Region: _______________ District: ___________________ Health facility: __________________ Location: ____________________________

Service delivery strategy: o fixed  o outreach  o mobile

| AGE RECOMMENDED | DOSE | 0–11 MONTHS | TOTAL | 12–23 MONTHS | TOTAL | 24 MONTHS OR OLDER | TOTAL | TOTAL VACCINATED |
|---|---|---|---|---|---|---|---|---|
| BIRTH | BCG | 00000 00000 00000 00000 00000 00000 00000 00000 | | 00000 00000 00000 00000 00000 00000 00000 00000 | | 00000 00000 00000 00000 00000 00000 00000 00000 | | |
| | OPV 0 | 00000 00000 00000 00000 00000 00000 00000 00000 | | ██████████████████ | | ██████████████████ | | |
| | Heb B BD <24 hours | 00000 00000 00000 00000 00000 00000 00000 00000 | | ██████████████████ | | ██████████████████ | | |
| 4 WEEKS | Heb B BD >24 hours | 00000 00000 00000 00000 00000 00000 00000 00000 | | 00000 00000 00000 00000 00000 00000 00000 00000 | | 00000 00000 00000 00000 00000 00000 00000 00000 | | |
| 6 WEEKS | OPV 1 | 00000 00000 00000 00000 00000 00000 00000 00000 | | 00000 00000 00000 00000 00000 00000 00000 00000 | | 00000 00000 00000 00000 00000 00000 00000 00000 | | |
| | IPV 1 | 00000 00000 00000 00000 00000 00000 00000 00000 | | 00000 00000 00000 00000 00000 00000 00000 00000 | | 00000 00000 00000 00000 00000 00000 00000 00000 | | |
| | PENTA 1 | 00000 00000 00000 00000 00000 00000 00000 00000 | | 00000 00000 00000 00000 00000 00000 00000 00000 | | 00000 00000 00000 00000 00000 00000 00000 00000 | | |
| | PCV 1 | 00000 00000 00000 00000 00000 00000 00000 00000 | | 00000 00000 00000 00000 00000 00000 00000 00000 | | 00000 00000 00000 00000 00000 00000 00000 00000 | | |
| | ROTA 1 | 00000 00000 00000 00000 00000 00000 00000 00000 | | 00000 00000 00000 00000 00000 00000 00000 00000 | | ██████████████████ | | |

[list of doses continues]

**Fig 3. Tally sheet design with doses grouped by recommended age with age segregated groups to mark age when dose is given.**

Date: _________ Region: _______________ District: ___________________ Health facility: __________________ Location: ____________________________

Service delivery strategy: o fixed  o outreach  o mobile

| # | Name of Child | Date of birth | Sex | Hep B | | BCG | | | Pentavalent (DTP-Hep B-Hib) | | | | | |
|---|---|---|---|---|---|---|---|---|---|---|---|---|---|---|
| | | | | | | | | | 1st | | | 2nd | | |
| | | | | <24 hours | >24 hours | Birth | <12 months | 1-4 years | 6 weeks | 12-23 months | 24-59 months | 10 weeks | 12-23 months | 24-59 months |
| | | __ / __ / __ | | | | | | | | | | | | |
| | | __ / __ / __ | | | | | | | | | | | | |
| | | __ / __ / __ | | | | | | | | | | | | |
| | | __ / __ / __ | | | | | | | | | | | | |
| | | __ / __ / __ | | | | | | | | | | | | |
| | | __ / __ / __ | | | | | | | | | | | | |
| | | __ / __ / __ | | | | | | | | | | | | |
| | | __ / __ / __ | | | | | | | | | | | | |
| | Total: | | | | | | | | | | | | | |

**Fig 4. Tally sheet design with doses associated with a child.**

Date was included in almost all tally sheets (n = 50, 98%) either at the top of the sheet indicating a month or specific date of the immunization activity. In register-style design (see Fig 4 in the examples above), date fields were multiple and linked to an individual visit and/or vaccines doses administered on specific dates.

Service delivery modality refers to whether the doses tallied were at a fixed site such as a health facility, at an outreach location, or by a mobile team. Service delivery modality was specified in 28 or 55% of tally sheets and not specified in 22 or 43%.

Service modality specifying whether doses were given at a fixed site, through outreach by health workers to a designated location in a community, or through mobile teams was observed most frequently in submissions from the African region (18 of 23 tally sheets from countries in this region), see Table 2.

## Child characteristics

More than three quarters of tally sheets (n = 42, 82%) have the possibility of capturing older aged children being vaccinated with delayed doses (also called catch-up), i.e., for age ranges above the recommended target age for vaccination according to the national schedule (see Table 3). Age segregation may be under or over a specific age, e.g., < 1 year, and >1 year, or have dose and age-related specifications, e.g., 0–11 months, 12–23 months, and 24–59 months. For example, if measles-containing vaccine second dose (MCV2) is recommended at 15 or 18 months, a tally sheet that permits recording of catch-up doses would include a space to record an MCV2 dose given at or after 24 months of age. Of the tally sheets with age segregation, 23 tally sheets did not have an upper age by which catch-up vaccine doses could be recorded (i.e., age column remained open, for example, 24m+), whereas 13 tally sheets indicated a closed age threshold such as 12–59 months, or specific to each dose (e.g., Rotavirus dose 1 given between 2–4 months and Rotavirus 2 given between 4–7 months). Nine tally sheets (18%) were restrictive by age to infants and young children with upper age thresholds at or below 24 months (Fig 5).

Other dimensions of disaggregation observed included sex, ethnicity, legal status, and specific population groups.

For sex-disaggregation, 20 tally sheets (39%) included data elements to mark whether the child is female or male, and 31 tally sheets (61%) did not specify.

Almost a quarter of tally sheets (n = 12 or 24%) had a field to mark whether a child was fully immunized. Three of these tally sheets specify fully immunized below 1 year of age; one tally sheet specified whether fully immunized by 2 years of age. One tally sheet has in line instructions to assess a child's immunization status, i.e., "Fully Immunized (FIC) after MR1. Verify if the child has received BCG, OPV 1-3, PCV 1-3, Penta 1-3, RVV 1-2, MR1 and Y/Fever vaccines according to schedule" (Fig 6).

## Vaccine and dose information

A few countries (n = 9, 18%) included tally fields related to newborn protection against tetanus. Among these, all have achieved neonatal tetanus elimination and none corresponded to the 10 countries where maternal and neonatal tetanus is still a public health problem [11]. Some countries include brief instructions, i.e., one tally sheet specifies: "verify from mother's Td [tetanus/diphtheria vaccine] record." Another tally sheet indicates to check children vaccinated for Penta1 [first dose of a pentavalent vaccine DTP-Hib-HepB] if protected at birth against neonatal tetanus, and a third tally sheet's in-line instruction for tetanus protection at birth is to: "tally a child who's [sic] mother has received 2 or more doses of Td doses during the last pregnancy."

For Hepatitis B birth dose timing, 14 (28%) tally sheets have separate fields for below 24 hours and above 24 hours. Among these countries, the upper age limit varies, e.g., two countries (>24 hours and up to 14 days or 2 weeks), one country (2–42 days), another example (>24 hours and up 11 months), and another (>24 hours and up to 11 months, 1–4 years, or 5–14 years).

For the upper age limit for rotavirus vaccination, 21 tally sheets (41%) had some designation: 11 indicated an upper age limit of 2 years, 9 indicated an upper age limit of 12 months, and one indicated an upper age limit of 8 months. The remaining countries either did not specify or do not have rotavirus in the national vaccination schedule.

**Table 2. Global eJRF tally sheet analysis, 2023 (data collected in 2024), occurrence of characteristics by WHO region for recording immunization data on country tally sheets.**

| | AFR n=23 | AMR n=12 | EMR n=5 | Others[1] n=11 | Total n=51 |
|---|---|---|---|---|---|
| *Site & session* | | | | | |
| **Location[2]** | | | | | |
| Yes | 19 (83) | 10 (83) | 5 (100) | 10 (91) | 44 (86) |
| No | 4 (17) | 2 (17) | - | 1 (9) | 7 (14) |
| **Date** | | | | | |
| Yes | 23 (100) | 12 (100) | 5 (100) | 10 (91) | 50 (98) |
| No | - | - | - | 1 (9) | 1 (2) |
| **Service delivery modality[3]** | | | | | |
| Yes | 18 (78) | 5 (42) | 2 (40) | 3 (27) | 28 (55) |
| No | 5 (22) | 6 (50) | 3 (60) | 8 (73) | 22 (43) |
| N/A | - | 1 (8) | - | - | 1 (2) |
| *Child* | | | | | |
| **Age groups segregated for catch-up (late) doses** | | | | | |
| Yes | 16 (70) | 8 (67) | 4 (80) | 10 (91) | 38 (75) |
| Partial[4] | - | 3 (25) | - | 1 (9) | 4 (8) |
| No | 7 (30) | 1 (8) | 1 (20) | - | 9 (18) |
| **Sex** | | | | | |
| Yes | 8 (35) | 3 (25) | 4 (80) | 5 (45) | 20 (39) |
| No | 15 (65) | 9 (75) | 1 (20) | 6 (55) | 31 (61) |
| **Fully immunized** | | | | | |
| Yes | 10 (43) | 1 (8) | - | 1 (9) | 12 (24) |
| No | 13 (57) | 11 (92) | 5 (100) | 10 (91) | 39 (76) |
| *Format* | | | | | |
| **Doses line listed** (not grouped by contact) | | | | | |
| Yes | 19 (83) | 10 (83) | 5 (100) | 10 (91) | 44 (86) |
| No | 4 (17) | 2 (17) | - | 1 (9) | 7 (14) |
| **Unique tally marks** (fill-in circle or tick box) | | | | | |
| Yes | 17 (74) | 2 (25) | 3 (60) | 1 (9) | 24 (47) |
| No | 6 (26) | 9 (75) | 2 (40) | 10 (91) | 27 (53) |
| **Subtotal count by dose by age** | | | | | |
| Yes | 17 (74) | 8 (67) | 4 (80) | 6 (55) | 35 (69) |
| No | 6 (26) | 4 (33) | 1 (20) | 4 (45) | 16 (31) |
| **Grand total count by dose** | | | | | |
| Yes | 7 (30) | 1 (8) | 3 (60) | 6 (55) | 17 (33) |
| No | 16 (70) | 11 (92) | 2 (40) | 4 (45) | 34 (67) |
| *Instructions* | | | | | |
| **Narrative on use and/or filled example** | | | | | |
| Yes | 4 (17) | 1 (8) | - | 3 (27) | 8 (16) |
| No | 19 (83) | 11 (92) | 5 (100) | 8 (73) | 43 (84) |

[1]Others category includes countries from EUR, SEAR and WPR

[2]Location refers to any write-in line for a health facility, administrative level (e.g., province/state/region, district, county, village)

[3]Service delivery modality refers to specifying whether the doses were given at a fixed site, through outreach and/or mobile teams

[4]Partial refers to tally sheets that have information to be able to calculate by date of birth and date of dose given

**Table 3. Global eJRF tally sheet analysis, 2023 (data collected in 2024), occurrence of characteristics by World Bank income classification for recording immunization data on country tally sheets by country income.**

| | Low income n = 9 | Lower middle income n = 24 | Upper middle income n = 12 | High income n = 6 | Total n = 51 |
|---|---|---|---|---|---|
| *Site & session* | | | | | |
| **Location[2]** | | | | | |
| Yes | 7 (78) | 21 (88) | 12 (100) | 4 (67) | 44 (86) |
| No | 2 (22) | 3 (13) | - | 2 (33) | 7 (14) |
| **Date** | | | | | |
| Yes | 9 (100) | 23 (96) | 12 (100) | 6 (100) | 50 (98) |
| No | - | 1 (4) | - | - | 1 (2) |
| **Service delivery modality[3]** | | | | | |
| Yes | 7 (78) | 16 (67) | 4 (33) | 1 (17) | 28 (55) |
| No | 2 (22) | 8 (33) | 8 (67) | 4 (67) | 22 (43) |
| N/A | - | - | - | 1 (17) | 1 (2) |
| *Child* | | | | | |
| **Age groups segregated for catch-up (late) doses** | | | | | |
| Yes | 7 (78) | 16 (67) | 10 (83) | 5 (83) | 38 (75) |
| Partial[4] | -- | 1 (4) | 2 (17) | 1 (17) | 4 (8) |
| No | 2 (22) | 7 (29) | - | - | 9 (18) |
| **Sex** | | | | | |
| Yes | 3 (33) | 10 (42) | 4 (33) | 3 (50) | 20 (39) |
| No | 6 (67) | 14 (58) | 8 (67) | 3 (50) | 31 (61) |
| **Fully immunized** | | | | | |
| Yes | 4 (44) | 8 (33) | - | - | 12 (24) |
| No | 5 (56) | 16 (67) | 12 (100) | 6 (100) | 39 (76) |
| *Format* | | | | | |
| **Doses line listed** (not grouped by contact) | | | | | |
| Yes | 7 (78) | 21 (88) | 11 (92) | 4 (67) | 43 (84) |
| No | 2 (22) | 3 (13) | 1 (8) | 2 (33) | 8 (16) |
| **Unique tally marks** (fill-in circle or tick box) | | | | | |
| Yes | 5 (56) | 15 (63) | 3 (25) | 1 (17) | 24 (47) |
| No | 4 (44) | 9 (38) | 9 (75) | 5 (83) | 27 (53) |
| **Subtotal count by dose by age** | | | | | |
| Yes | 6 (67) | 17 (71) | 8 (67) | 4 (67) | 35 (69) |
| No | 3 (33) | 7 (29) | 4 (33) | 2 (33) | 16 (31) |
| **Grand total count by dose** | | | | | |
| Yes | 3 (33) | 9 (38) | 2 (17) | 3 (50) | 17 (33) |
| No | 6 (67) | 15 (63) | 10 (83) | 3 (50) | 34 (67) |
| *Instructions* | | | | | |
| **Narrative on use and/or filled example** | | | | | |
| Yes | 2 (22) | 5 (21) | 1 (8) | - | 8 (16) |
| No | 7 (78) | 19 (79) | 11 (92) | 6 (100) | 43 (84) |

[1]Others category includes countries from EUR, SEAR and WPR

[2]Location refers to any write-in line for a health facility, administrative level (e.g., province/state/region, district, county, village)

[3]Service delivery modality refers to specifying whether the doses were given at a fixed site, through outreach and/or mobile teams

[4]Partial refers to tally sheets that have information to be able to calculate by date of birth and date of dose given

**Recording late doses, <u>open</u>** (n=25)

| 0–11m | 12–23m | 24m+ |
|---|---|---|

3 groups, open: **4 countries**

| 0–11m | 12m+ |
|---|---|

2 groups: **10 countries**

| 0–11m | "outside target" |
|---|---|

2 groups: **1 country**

Other groupings: **11 countries**

- - - - - - - - - - - - - - - - - - - - - - - - - - - - - - - - - - - - - - - - - - - - - - - - - -

**Recording late doses, <u>closed</u>** (n=13)

| 0–11m | 12–23m | 24–59m |
|---|---|---|

3 groups: **3 countries**

| 0–11m | 12–59m |
|---|---|

2 groups: **4 countries**

Other groupings closed <4, 5 or 6y: **6 countries**

- - - - - - - - - - - - - - - - - - - - - - - - - - - - - - - - - - - - - - - - - - - - - - - - - -

**No groupings** (relies on date dose given & date of birth) (n=4)

- - - - - - - - - - - - - - - - - - - - - - - - - - - - - - - - - - - - - - - - - - - - - - - - - -

**Recording late doses, <u>restrictive</u>** (n=9)

| 0–11m | 12–23m |
|---|---|

2 groups: **7 countries**

| 0–23m |
|---|

1 groups: **1 country**

Other groupings <18m: **1 country**

**Fig 5. Age groupings for recording late doses.**

### Format

A majority of tally sheets line list doses (n = 43, 84%), typically vertically, while the remaining tally sheets group by contact (n = 6, 16%).

Almost half of tally sheets (n = 24, 47%) have an explicit way of marking a dose given, i.e., with a fill-in circle or box. More than half of tally sheets (n = 27, 53%) have a blank field which allows for different ways of marking, i.e., hash marks, numbers. etc.

Subtotal and grand totals are seen with different data elements, e.g., when there is age and/or sex segregation for specific doses. Subtotals and grand totals provide fields to summarize values on a tally sheet for other recording tools, e.g., monthly reports. 35 (69%) of tally sheets had subtotals, and 17 (33%) had grand totals (see Fig 2). Subtotals may be for age of dose given or by sex (see Fig 7). Grand totals fields exist in 17 (33%) of tally sheets.

### Instructions

Few tally sheets included basic instructions and/or examples of how to fill a tally sheet (n = 8, 16%) whereas the rest (n = 43, 84%) had no instructions. Examples of instructions ranged from basic, i.e., one country in the AFRO region: "*Pour chaque vaccin administré à l'enfant barrer un cercle correspondant à l'âge et à l'antigène concerné. Le total pour chaque antigène d'après la tranche d'âge sera reporté dans le formulaire n° 1 PEV du mois en cours*" [For each vaccine

| STATUS | UNDER 1 YEAR OF AGE | TOTAL | UNDER 2 YEARS OF AGE | TOTAL |
|---|---|---|---|---|
| FULLY IMMUNIZED BY 12 MONTHS | ooooo ooooo ooooo ooooo ooooo ooooo ooooo ooooo ooooo ooooo ooooo ooooo<br><br>Verify the child received BCG, OPV1–3, PCV 1–3, Penta 1–3, RVV 1–2, IPV 1–2, MR1 and Yellow fever vaccines according to schedule | | ████████████ | |
| FULLY IMMUNIZED BY 24 MONTHS | ████████████ | | ooooo ooooo ooooo ooooo ooooo ooooo ooooo ooooo ooooo ooooo ooooo ooooo ooooo ooooo<br><br>Verify the child received MR2 and all other vaccines according to schedule | |

BCG Bacillus Calmette-Guérin
DTP Diphtheria-tetanus-pertussis vaccine
Hep B Hepatitis B vaccine
Hib *Haemophilus influenzae* type b vaccine
IPV Inactivated poliovirus vaccine
MR Measles rubella
OPV Oral poliovirus vaccine
Penta Pentavalent combination vaccine against diphtheria, tetanus, pertussis, Hep B, Hib
PCV Pneumococcal conjugate vaccine

**Fig 6. Example tally sheet fields for indicating fully vaccinated by two age groupings.**

**Fig 7. Example of tally sheet with fields for sex, age groups, sub-totals, and totals.**

[list of doses continues]

BCG Bacillus Calmette-Guérin
BD birth dose
Hep B Hepatitis B vaccine
OPV Oral poliovirus vaccine

administered to the child, cross out a circle corresponding to the age and the antigen concerned. The total for each vaccine does according to the age group will be reported on EPI form no. 1 for the current month], to more detailed including visual diagrams.

## Discussion

Tally sheets are unequivocally an essential tool for health workers and programme managers. Especially in low-resource settings, the tally sheet is the primary data recording and reporting instrument for health workers to mark vaccine doses administered alongside the register book [12,13]. As such, the tally sheet is the main source of administrative data at the vaccination delivery point – the lowest level or unit of analysis – and therefore foundational to the immunization data system in almost all countries. Several factors including design aspects presented in this analysis [14] directly influence the accuracy of how doses are recorded and reported to higher levels of the health system. For these reasons, while tally sheets are primarily a tool used by health workers, their design and periodic updating is of interest to stakeholders including users of administrative immunization data for statistics, policy and programming purposes.

This first global analysis, though limited by only about a quarter of reporting entities submitting documents retained, shows heterogeneity in immunization tally sheet design and variation in the extent that sampled tally sheets are aligned with globally recommended elements found in recent vaccine-specific and programmatic guidance related to catch-up vaccination.

Data elements in the tally sheets reviewed included site and session characteristics, doses according to the national vaccine schedule, and programmatic priorities. Date and location provide a timeframe and area for which vaccine doses are given and tallied. Having service delivery strategy denoted – e.g., at a health facility fixed site, outreach, and mobile – captures data on the service modality by which children were reached, i.e., through care-seeking to a fixed site or by health workers bringing immunization services closer to communities. This service delivery data compiled across tally sheets can inform catchment area microplanning by considering proportion of infants and children in the catchment area reached by each service and refine resource allocation and scheduling [15].

Timely vaccination is important to ensure that populations are protected against vaccine-preventable diseases as early as possible, however, vaccinations may be missed for a wide variety of reasons leaving individuals vulnerable unless caught up. Restrictive data recording and reporting tools has been often cited as a barrier for health workers to administer delayed vaccination to children who are missing doses and contributes to missed opportunities for vaccination [16]. Catch-up vaccination has gained momentum globally with the *Big Catch-up* and countries have updated policies, schedules, and recording and reporting tools to reach older aged children who missed routine vaccinations [17,18]. Age segregation is critical to distinguish between "on-time" vaccinations given and doses caught up among older children missing recommended doses. Tally sheets should be designed in a way that guides health workers to accurately record all vaccination doses, in line with the national catch-up policy and schedule [4]. For Hepatitis B birth dose, it is recommended for the tally sheet to distinguish between "timely" doses given <24 hours – the main indicator for coverage – and more than 24 hours after birth [3]. Rotavirus vaccine doses have a recommended upper age threshold [19] seen in some of the tally sheet examples reviewed. MCV2 introduction experiences [20], and for all new vaccines introduced such as malaria vaccine with multiple doses [21], require appropriate age segregation in the tally sheet design.

Gender inequality is a known predictor of childhood immunization coverage [22] but differences in coverage by sex, which is systematically monitored by immunization coverage surveys and electronic immunization registries where they exist, are generally not significant at the national level [23]. Sex segregation of doses administered, as seen in some tally sheets in this analysis, can further assist in identifying and understanding inequities in specific subnational areas, however, in order for this data to be actionable, reliable sex-specific denominators are needed. Sex disaggregation adds more data fields and variables so should be considered in relation to programme objectives, and with practical considerations like space on the page and font size as a factor.

Ease of use of a tally sheet relates to the layout and clear designated ways to mark or tick, such as fill-in circles or boxes. To assist in reducing arithmetic errors during monthly data compilation [24], the format should be pre-tested and intuitive, and benefits from including sub-totals and grand totals to assist in aggregating as well as looking back to try to identify the source or location of discrepancies.

Instructions can provide some clarity for filling the tally sheet through an explanatory page, example filled format, and/or in-line brief definitions or reminders. Tally sheets are complementary to other immunization recording and reporting tools such as registers, monthly reports and home-based records. Review and revision of these tools should be done concurrently and with the appropriate definitions and instructional elements. Written tally sheet instructions alone are insufficient for good tally practice and require reinforcing through checking, performance feedback, supportive supervision, and other mechanisms [25].

Immunization data quality in low and middle-income countries, characterized as varied, can be improved by appropriate data collection tools, increasing health workers' capacities and motivation through training, motivation, and feedback [26,27]. Digitizing health worker tools through person-centered point of service systems is promising [28,29]. Reliance on paper-based methods at the point of delivery will continue in many countries in the meantime. Thus, efforts should be made to improve these tools and make them more user-centered. Human-centered design has become more prominent in health care in recent years, including in supporting preventative services in low and middle-income settings [30,31], and this approach could be considered when revising tally sheets. Electronic immunization registries could eventually help reduce some of the challenges with paper systems, but feasibility needs to be assessed before completely replacing basic tools [32].

Even when the tally sheet is well designed, it needs to be available at all service delivery points. Stockouts of tally sheets occur in many countries [33,34] similar to stockouts of home-based records [35], and relate to different factors such as changes in immunization schedules requiring updates to these records, with associated costs, and unclear responsibilities for printing and distribution [36].

## Strengths and limitations

This study has several strengths. First, it is the first global analysis of tally sheet design as reported through the eJRF, illustrating how this foundational tool for monitoring immunization programmes varies and may or may not facilitate data recording and subsequent aggregation. Second, it presents tally sheet data elements and important considerations at a time when infant and child immunization schedules include more vaccine doses and policy revisions allow for catch-up vaccination in older aged children who missed on-time vaccination. Third, it emphasizes the limited evaluation of these tools' design and usability and thus encourages implementation research in this area, as recently done for home-based records.

There are also several limitations to this analysis. As not all countries shared their tally sheets, the sample of tally sheets reviewed is not representative and the findings cannot be generalized. Tally sheet review was restricted to the submitted document; a country may have a tally sheet example, though if it was not submitted or a document not meeting inclusion criteria was submitted in its place, it is not part of the analysis. Varying quality of some tally sheets (e.g., partial screenshot) led to exclusion of one tally sheet. The review does not include associated materials, i.e., if there are accompanying materials, i.e., a cover page with location information, or instructions on how to use a country's tally sheet that were not submitted. Finally, this analysis focuses on tally sheet design and does not assess how tally sheets are used in practice alongside other recording and reporting tools.

## Conclusions

As national immunization programmes and vaccination schedules protect against more diseases, and thereby become more complex, support mechanisms to health workers including fit-for-purpose tally sheet design and instruction, and

associated supportive supervision and feedback are essential [37], similar to what is recommended for home-based records.

## Supporting information

**S1 Data. Data extracted from retained tally sheets submitted to eJRF 2023 (data collected in 2024).** (XLSX)

## Acknowledgments

We are grateful to staff in Ministries of Health who submitted an example of their tally sheet included in this analysis. We also acknowledge Dr. Sylvester Maleghemi, WHO Headquarters Department of Immunization, Vaccines and Biologicals, for his useful review and feedback on the manuscript.

   **Ethics and consent:** No individual-level data were used in this study. Only country-level data officially reported by Ministries of Health to WHO and UNICEF were used. As a result, ethical approval and consent was not required.

## Author contributions

**Conceptualization:** Ariel Higgins-Steele, Marta Gacic-Dobo, M. Carolina Danovaro-Holliday.

**Data curation:** Ariel Higgins-Steele, M. Carolina Danovaro-Holliday.

**Formal analysis:** Ariel Higgins-Steele.

**Methodology:** Ariel Higgins-Steele, Stephanie Shendale, M. Carolina Danovaro-Holliday.

**Supervision:** M. Carolina Danovaro-Holliday.

**Validation:** Stephanie Shendale, Jan Grevendonk, Marta Gacic-Dobo, M. Carolina Danovaro-Holliday.

**Visualization:** Ariel Higgins-Steele.

**Writing – original draft:** Ariel Higgins-Steele.

**Writing – review & editing:** Stephanie Shendale, Jan Grevendonk, Marta Gacic-Dobo, M. Carolina Danovaro-Holliday.

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
