## [Decision Letter · Decision Letter 0]

15 Aug 2025

PGPH-D-25-01834

Recording vaccine doses administered: A global analysis of tally sheet design for infant and child immunizations

Dear Dr. Higgins-Steele,

Thank you for submitting your manuscript to PLOS Global Public Health. After careful consideration, we feel that it has merit but does not fully meet PLOS Global Public Health’s publication criteria as it currently stands. Therefore, we invite you to submit a revised version of the manuscript that addresses the points raised during the review process.

We look forward to receiving your revised manuscript.

Kind regards,

Paulo Jorge Gonçalves de Bettencourt, Ph.D.

Academic Editor

Journal Requirements:

1. We have amended your Competing Interest statement to comply with journal style. We kindly ask that you double check the statement and let us know if anything is incorrect.

2. Please upload a copy of Figures 1 to 5 which you refer to in your text on pages 5, 7, 13, 14 and 16. Or, if the figure is no longer to be included as part of the submission please remove all reference to it within the text.

Additional Editor Comments (if provided):

Reviewers' comments:

Reviewer's Responses to Questions

**Comments to the Author**

1. Does this manuscript meet PLOS Global Public Health’s publication criteria?

Reviewer #1: Yes

Reviewer #2: Yes

2. Has the statistical analysis been performed appropriately and rigorously?

Reviewer #1: Yes

Reviewer #2: Yes

3. Have the authors made all data underlying the findings in their manuscript fully available (please refer to the Data Availability Statement at the start of the manuscript PDF file)?

Reviewer #1: Yes

Reviewer #2: Yes

4. Is the manuscript presented in an intelligible fashion and written in standard English?

Reviewer #1: Yes

Reviewer #2: Yes

Reviewer #1: Firstly, thank you very much for providing the opportunity to review this manuscript. This study is well-conducted, and the researchers are commended for their work thorough evaluating Recording vaccine doses administered: A global analysis of tally sheet design for infant and child immunizations. The study is effectively structured, with a clear identification of the impact of tally sheet design for infant and child immunizations in different countries. However, to enhance its credibility and ensure clear communication of the findings, the researchers should address the minor comments and suggestions provided above. Once these revisions are made, the study will be ready for publication and will provide valuable insights for infant and child immunizations.

Reviewer #2: Dear authors,

The presented research article "Recording vaccine doses administered: A global analysis of tally sheet design for infant and child immunizations" contributes original research closely related to current literature. It addresses one of contemporary main challenges in immunisation, particularly in low-income countries, i.e., immunisation data quality. Accuracy and completeness of these data are essential for vaccination coverage monitoring, identifying missed opportunities, following up progress, and guiding correction measures for vaccination teams.

The study utilises open international data and qualitative and quantitative analysis. Used data processing algorithm with reproducibility has been made available by researchers, and additional tables are given. Results obtained are valuable and useful, specifically because data sources come from around the globe. However, since completeness in data varied significantly from one country to another, making robust generalizable inferences is not possible. Despite this limitation, the observations remain valuable for future work as well as in policy debates.

The article has not been published previously and it has been posted only at a preprint server, which is in line with PLOS's policy.

Specific recommended adjustments:

• Line 159: Change C to “See.”

• Line 233: Change “who’s” for "whose"

• Line 233: Define sic before first use.

• Line 302 and 306: Change existing quotation ' ' marks to “” to maintain consistency.

• Line 342: keep the consistent hyphenation of “stock-outs” versus “stockouts.”

**Do you want your identity to be public for this peer review?** For information about this choice, including consent withdrawal, please see our Privacy Policy

Reviewer #1: **Yes: ** Full name: Habtamu Molla Ayele

First name: Habtamu Molla

Last name: Ayele

Reviewer #2: No

---

## [Editor Report · Decision Letter 1]

23 Sep 2025

Recording vaccine doses administered: A global analysis of tally sheet design for infant and child immunizations

PGPH-D-25-01834R1

Dear Ms Higgins-Steele,

We are pleased to inform you that your manuscript 'Recording vaccine doses administered: A global analysis of tally sheet design for infant and child immunizations' has been provisionally accepted for publication in PLOS Global Public Health.

Best regards,

Paulo Jorge Gonçalves de Bettencourt, Ph.D.

Academic Editor